# Machine Learning in Colorectal Cancer Risk Prediction from Routinely Collected Data: A Review

**DOI:** 10.3390/diagnostics13020301

**Published:** 2023-01-13

**Authors:** Bruce Burnett, Shang-Ming Zhou, Sinead Brophy, Phil Davies, Paul Ellis, Jonathan Kennedy, Amrita Bandyopadhyay, Michael Parker, Ronan A. Lyons

**Affiliations:** 1Swansea University Medical School, Swansea SA2 8PP, UK; 2Faculty of Health, University of Plymouth, Plymouth PL4 8AA, UK; 3Clinithink Ltd., Bridgend CF31 1LH, UK

**Keywords:** machine learning, colorectal cancer, risk prediction, scoping review

## Abstract

The inclusion of machine-learning-derived models in systematic reviews of risk prediction models for colorectal cancer is rare. Whilst such reviews have highlighted methodological issues and limited performance of the models included, it is unclear why machine-learning-derived models are absent and whether such models suffer similar methodological problems. This scoping review aims to identify machine-learning models, assess their methodology, and compare their performance with that found in previous reviews. A literature search of four databases was performed for colorectal cancer prediction and prognosis model publications that included at least one machine-learning model. A total of 14 publications were identified for inclusion in the scoping review. Data was extracted using an adapted CHARM checklist against which the models were benchmarked. The review found similar methodological problems with machine-learning models to that observed in systematic reviews for non-machine-learning models, although model performance was better. The inclusion of machine-learning models in systematic reviews is required, as they offer improved performance despite similar methodological omissions; however, to achieve this the methodological issues that affect many prediction models need to be addressed.

## 1. Introduction

Colorectal cancer (CRC) is the third most common cancer worldwide, affecting both males and females to a similar extent, with a lifetime risk of around 5% [1]. Whilst the survival rate for CRC has improved significantly since the early 1970s [2], the number of cases in the United Kingdom is expected to increase by up to 30% by 2035 [2,3]. Several screening modalities are used in practice; all have been shown to prevent CRC deaths, with colonoscopy considered the most effective method of screening [4]. Current screening strategies are determined primarily by age [5] and family history [6]. There are harms associated with colorectal screening [7], which has led to interest in targeted risk-based screening. Risk-based screening offers a means of minimising harm to most of the population who are at low risk, whilst identifying those at higher risk for targeted screening [8].

Globally the uptake of colorectal screening programmes varies, but is generally less than 65% [9]. The UK has failed to reach a 75% target despite attempts to improve uptake of the colorectal screening programme [10]. Only 10% of cases are detected by screening and there is an increasing number of younger (age < 50 years) individuals presenting symptomatically [11]. This is another driver for targeted colorectal screening based on risk stratification [12].

Risk-prediction models for cancer, including CRC are available in clinical practice both here in the UK [13,14,15,16] and elsewhere [17,18], but translating prediction models into clinical practice is challenging [15]. The low adoption rate of such techniques in clinical practice could be considered as contributing to research waste [19]. Additionally, new features to be considered in prediction models are regularly identified and the volume and breadth of collected data continues to increase. Prediction models that include genetic information, such as selected single nucleotide polymorphisms [20], environmental factors [21], and lifestyle information [22], are being developed and these features are likely to become routinely collected data in the coming years.

Multivariable prediction models utilise two or more variables (predictors, features) to estimate the risk or probability of an event being experienced by an individual, such as diagnosis of a condition or a condition-related outcome (prognosis) [23]. Machine-learning (ML) models are considered to be computation intensive, data-driven approaches, with fewer modeller decisions and more powerful prediction capability than traditionally required [24].

To date, there have been many CRC risk-prediction models published pertaining to unique populations and care settings and which incorporate a range of risk factors. Several recent systematic reviews of CRC risk-prediction models have been undertaken [25,26,27,28], which have identified common limitations in risk-prediction model publications, including:Lack of external validationDifferences in how factors are measured, e.g., blood pressureManagement of missing data related to these factorsLack of calibration curves

These systematic reviews have identified a wide range of factors that are considered for inclusion in prediction models, as well as variation in the definition of a case. Apart from a review by Grigore et al. [28], none of these systematic reviews have included models developed using machine-learning techniques and reported only models developed using traditional statistical techniques.

Machine-learning techniques can make use of historically collected, stored data to create predictions for future populations or individuals. This includes determining those at higher risk of developing cancer [29]. Two reviews of machine learning and cancer risk [30,31] identified breast-cancer models as the most reported type of cancer-risk prediction model, with reported accuracy most commonly around 70%. Both noted a lack of appropriate publications for inclusion and identified poor or absent methodology in testing or validating models and increasing data dimensionality. The increasing dimensionality and the noisiness (e.g., presence of unimportant factors, missing data or gaps in the data, outliers, duplicate or inaccurate data) of such health data are well-recognised [32,33], with appreciation of the varied penetrance of factors across a population [34].

The lack of machine-learning models in these reviews provides a reason for assessing the potential value of models using this approach. Understanding how and why such models are rarely included in systematic reviews of CRC risk-prediction models can be used to inform better reporting in publications. It is important to identify methodologies, risk factors and limitations that can be used to inform the development of CRC risk-prediction models that can be included in future systematic reviews. However, the promise of machine-learning approaches in comparison to traditional methods for tackling these problems is not without debate [35].

This literature review focuses on machine-learning derived colorectal-cancer-risk-prediction models that utilise routinely collected health data, which are likely to be held in health databanks. We review their strengths and weaknesses, including their overall performance.

## 2. Materials and Methods

We undertook a comparison and assessment of machine-learning models for the risk prediction of colorectal cancer (colon or rectum) or colon cancer alone. A scoping review approach was used as it was unclear how many publications or models would be identified. We felt there was likely to be considerable heterogeneity in the available literature, in terms of the methodologies, variables included and the definition of routinely collected data relating to colorectal cancer risk and prognostic models. This scoping review approach provided an opportunity to identify gaps in knowledge about the literature around ML-colorectal-cancer-prediction models, as well as determining if a systematic review is warranted [36].

### 2.1. Eligibility Criteria

The inclusion criteria were: full-text availability; English language; articles related to colorectal or colon cancer; published between 2010 and 2019 inclusive; with explanatory variables derived from routinely collected demographic and health data. Articles were excluded if only rectal cancer was reported, only image assessment was included, or if they were based on genetic material assessment. The inclusion and exclusion criteria were defined based on whether the articles focused on colorectal or colon cancer. Date limits were initially set for the period 1 January 2010 to 31 December 2019 and extended to 31 December 2020 as being pertinent to the availability of routinely collected data and the increasing interest in the use of machine learning. A scoping review approach was used to ensure broad consideration of models, included features and identification methods. The search was not restricted to certain types of features or a maximum number of features to maximise the opportunity to explore the diversity of features considered and methodologies applied.

### 2.2. Inclusion Criteria for Publications

Publications were selected for review if they:used one or more machine-learning methods, including comparison with logistic regressionused at least two risk factorsused an experimental or observational study design

### 2.3. Exclusion Criteria for Publications

Publications were considered ineligible for inclusion if they:used images or image parameters as risk factorsincluded genetic informationincluded molecular markersutilised questionnaire-derived data, e.g., nutrition questionnaireswere designed to test performance of a risk factor for colorectal cancerwere review articleswere conference abstracts

### 2.4. Information Sources

Four databases were searched: Cochrane Library, PubMed, Web of Science and IEEE.

### 2.5. Search Strategy for PubMed (L3 Heading)

The search strategy related to a number of concepts: colorectal cancer, colon cancer, prediction models (statistical models) and machine learning using a combination of MeSH terms and free-text terms. These were combined to form the following search string:

Search: (((“Risk”[Mesh] OR risk[tw] OR hazard[tw] OR likelihood[tw] OR probabil*[tw]) OR (“Prognosis”[Mesh] OR prognosi*[tw] OR prognosti*[tw])) AND ((“Colorectal Neoplasms”[Mesh] OR ((colorectal[tw] OR colorect*[tw]) AND (tumo*[tw] OR cancer[tw] OR carcinom*[tw] OR neoplas*[tw] OR malignan*[tw]))) OR (“Colonic Neoplasms”[Mesh] OR ((colon[tw] OR bowel[tw] OR colon*[tw]) AND (neoplas*[tw] OR tumo*[tw] OR cancer[tw] OR carcinom*[tw] OR malignan*[tw]))))) AND (((“Models, Statistical”[Mesh] OR “ROC Curve”[MESH] OR “predict* tool*”[tw] OR nomogram*[tw] OR “predict* model*”[tw] OR decision*[tw] OR scor*[tw] OR algorithm*[tw] OR “risk scor*”[tw] OR “sensitivity and specificity*”[tw] OR sensitivity[tw] OR specificity[tw] OR “predictive value of tests”[tw] OR prediction*[tw] OR “receiver operating characteristic curve*”[tw] OR “ROC curve*”[tw] OR “area under curve*”[tw] OR “area under curve”[tw] OR “area under the curve*”[tw] OR AUC[tw] OR “C statistic*”[tw] OR discriminat*[tw] OR classif*[tw] OR “absolute risk*”[tw] OR brier*[tw] OR “lorenz curves”[tw] OR calibration[tw] OR indices[tw] OR stratify*[tw] OR “c-statistic”[tw] OR “C statistic”[tw] OR “statistical learning”[tw] OR “statistical-learning”[tw] OR “positive predictive value*”[tw] OR “negative predictive value*”[tw] OR AUROC[tw] OR “c-index”[tw] OR concordance[tw] OR DCA[tw])) OR ((“Machine Learning”[Mesh] OR “neural network*”[tw] OR “decision tree*”[tw] OR “support vector machine*”[tw] OR SVM[tw] OR “random forest*”[tw] OR “naive bayes” [tw] OR “machine learn*”[tw] OR “machine model*”[tw] OR “Artificial Intelligence”[Mesh] OR “Deep Learning”[Mesh] OR “Supervised Machine Learning”[Mesh] OR “supervised machine”[tw] OR “supervised learn*”[tw] OR “support vector machine”[tw] OR “relevance vector machine*”[ tw] OR “multi* layer perceptron*”[ tw] OR “RF classif*”[ tw] OR “bayes* network*”[tw] OR “nearest neighb*”[tw] OR KNN[tw] OR ANN[tw] OR RNN[tw] OR RF[tw] OR NB[tw] OR CART[tw] OR DT[tw] OR MLP[tw] OR “elastic net”[tw] OR BBN[tw] OR “deep learn*”[tw])))—Saved search Filters: from 2010/1/1–2019/12/31 Sort by: Most Recent.

These searches returned 20,211 articles from PubMed, 858 from the Web of Science, 1124 from IEEE and 199 from the Cochrane database.

The selection process was undertaken by a single author (BB) who selected articles based on the inclusion and exclusion criteria applied to the abstract and title. Where a decision could not be made on the abstract and title alone, the full article was considered. The selection of risk prediction and prognosis studies was undertaken concurrently as part of a larger project. Prognostic models were not considered in the review.

### 2.6. Data Charting

Data items were extracted from the fourteen selected articles using the CHARMS checklist [37] for assessment of applicability and the risk of bias by a single author (BB). The extracted data were recorded in a bespoke Excel spreadsheet. Data extraction was validated by the other authors (AB, SB, JK, SZ and MP). Disagreements and clarifications were discussed between the reviewers. No further adjudication was required. The data charting summary is provided as Appendix A, Appendix A.

Critical appraisal was based on data assessment with reference to the CHARMS checklist and risk of bias assessment using the PROBAST tool [38,39].

## 3. Results

### 3.1. Article Selection

The search terms focused on methodology relating to machine learning. A broad inclusion of terms was used to avoid missing potential articles and included both risk prediction and prognostic algorithms. In the absence of a definition of “routinely collected data”, further limits to the searches were not possible and relied on manual search of titles and abstracts. The PubMed search retrieved 20,350 publications for assessment, the Web of Science retrieved 1490 publications for assessment, and the Institute of Electrical and Electronic Engineers (IEEE) database retrieved 1124 publications. A total of 199 publications were retrieved from the Cochrane database. An initial manual review of the abstracts and titles excluded 21,547 articles and identified 1620 for further screening. Further screening involved manually removing remaining articles that were not relevant to the inclusion criteria. This left 65 reports that were assessed in detail, thirty of which were removed because of inclusion of omics or biomarker variables, use of natural language processing, utilisation of locally developed questionnaires or missing information. An additional eight reports were identified from the references of the remaining 65 reports and assessed for inclusion. Three were excluded for not meeting the inclusion and exclusion criteria. This provided 40 relevant reports of which 14 were related to risk-prediction models, the remainder being prognostic models. The 14 reports that included risk-prediction models were included in the scoping review.

A PRISMA flow diagram is provided as Figure 1.

### 3.2. Model Descriptions

The approaches undertaken are summarised in Table 1.

### 3.3. Model Purpose and Population Description

The purpose of all the models was related to colorectal cancer, except for one which considered only colon cancer [48]. A summary of the population for each model is provided in Table 2.

### 3.4. Model Performance

The assessment of model performance utilised a range of measures and was not consistently reported across the included studies; it is summarised in Table 3. The most frequently reported measure was sensitivity, with accuracy only reported in one study [40]. Values were not always reported in the article text; access to Appendix A was required in some instances and was not always accessible. Corresponding authors provided some missing Appendix A, but this was not always complete. The confidence intervals for model performance measures were not always reported.

### 3.5. Model Comparison and Benchmarking with CHARM Criteria

The review identified fourteen prediction models that utilised ML techniques for CRC prediction. Whilst examples of prediction models that included ML methods were limited, they provided some interesting results.

The search retrieved several validation studies for one ML model included in the review [42]. All models utilised some form of registry data. Consecutive recruitment could not be confirmed but was unlikely; however, for one of the single-institution-derived models, recruitment was via a clinic and patients were followed prospectively, which was unique amongst the articles reviewed [41]. The type of data source registry varied, ranging from national health questionnaires [52,54] to screening hubs [51]. Combining different datasets, such as the addition of a diabetes dataset [43] or multi-country datasets [42], during development and external validation was also observed. Indeed, only one ML model included both development and validation in the same study [42]. All the ML models provided information on the inclusion and exclusion criteria, apart from one study that was only available as an abstract [54]. The data collection periods covered the 1990s and 2000s for a variety of time periods, the shortest being the prospective study of Wu et al. [41].

There was also variety in the outcomes for the studies, both in terms of the period over which prediction was made to the outcome measure, with a range from cancer development, high-risk polyps, cancer risk following colonoscopy or based on direct screening decisions. These differences defined the variation in the variables considered for inclusion in the various models. Comorbidity and symptoms were frequently included; all ML models included age and gender. The number of variables included in the final model was also diverse, ranging up to 20 variables where reported, although actual numbers were rarely provided, and selection was inherent to the ML technique, such as random forest or neural networks. Determination of which variables to include in the model used different approaches, such as univariate and multivariate regression to ML methods. Some feature selection was part of the model development process, such as that observed with the tree-based methods. The actual number of variables used was not always reported, so, determining the event per variable ratio was not always possible. The suggested minimum of 100 events and 100 non-events [55] or an event-per-variable rate minimum of 10 [56] were not always possible to determine [40] and were not always met [41,44].

Information on how much missing data was present and how this was managed was frequently not described. A range of techniques for managing missing data was reported. Exclusion was a common first step, followed by imputation techniques, whilst the prospective study relied on scrutiny of records to complete missing or suspicious values. Hornbrook et al. [45] employed substitution of controls where data was missing.

A range of performance measures was reported. Importantly, none of the measures relied solely on an AUC or c-index. Comprehensive performance measure reporting was common. The best reported models were those from Kinar et al. [42] and Nartowt et al. [52], which included calibration plots. Odds ratios were frequently included in validation studies as part of performance measurement. Nomograms were not present in any of these publications.

Two authors compared multiple ML techniques [40,50]. These comparisons resulted in the best performance being attributed to neural networks. Of the remaining publications, eight used tree-based models and four used neural networks. Of the neural networks, the model proposed by Kinar et al. [42] was validated in multiple populations [45,46,47,49], including adaptation by inclusion of additional blood test results [48]. Interestingly one study utilised the same variables within a logistic regression model and achieved similar performance [48]. The studies using an approach of combining age, gender and blood counts utilised varying sizes of population dataset. The initial development model was trained on 466,107 patients. Of these, 2437 were diagnosed with CRC (control-to-case ratio of approximately 190:1), whilst the test set had a control-to-case rate of 200:1; with external validation, this was reduced to 5:1. Later validation and expansion studies for this model utilised control-to-case ratios ranging from 18:1 [45] up to 850:1 [47]. Hornbrook et al. [45] validated this on US data with 10 controls per case. Hilsden et al. [49] screening only individuals in Canada had a ratio of 17 controls per case. The expansion of the algorithm by Goshen et al. [48] used a ratio of 30 controls per case. With the exception of the Hilsden et al. [49] article all the studies had colorectal cancer as their outcome. Beyond this, the reported models had populations with a wide range of control-to-case ratios, from 3:1 [41,51] up to 400:1 [52]. The majority utilised a significantly imbalanced control-to-case ratio, but this was more representative of the colorectal cancer prevalence within a population of around 1%, equivalent to 100 controls per case.

Of the remaining tree-based models, Wu et al. [41] adopted a prospective approach, collecting data from patients referred for colonoscopy rather than from an asymptomatic population, with colorectal cancer diagnosis as the outcome measure. This prospective approach limited the population size, particularly as this was a single tertiary referral (specialised) institution, potentially accounting for the higher-than-average event rate, reflecting the non-standard risk pattern of the population.

Kop et al. [50] adopted a previously published approach [57] and compared a random forest and classification and regression tree (CART) approach with logistic regression, again using colorectal cancer diagnosis as the outcome measure. They also discussed the potential for using the change in values or rate of change as a variable, but decided not to include this due to the increased granularity of the data, a significant potential issue that may have a negative impact due to reduction in the population size from a lack of multiple recordings. The authors also recognised the potential for missing or incorrect data, including that relating to the CRC diagnosis. Interestingly, they used age and gender as their simplest comparison model and achieved comparable performance with more complex models, including logistic regression and machine-learning approaches.

The four remaining models used neural networks. Nartowt et al. [52] used a questionnaire-sourced dataset [58] covering a twenty-year period, with 583,770 controls and 1409 cases, with diagnosis within four years of the survey date and around 400 controls per case. This approach is interesting because it does not require any invasive techniques to collect data. The output risk score (between 0 and 1) was categorised into one of three risk levels rather than the low- and high-risk levels proposed by the US screening guidelines [59]. These authors compared the performance of their neural network with a range of screening methods, including colonoscopy and FIT, against which, unsurprisingly, it compared unfavourably. Although this review did not include screening ML algorithms, such as those proposed by Semmler et al. [60], it is important to recognise that ML approaches are frequently criticised when considered as an alternative to diagnostic screening, primarily because it is not understood which features distinguish cancers that require intervention and those that do not.

The neural network developed by Cooper et al. [51] utilised data from a UK bowel cancer screening programme, from two hubs over a six-month period during 2014. The cases included both CRC and high-risk adenomas. The study was designed to compare two different screening methods: faecal occult blood test and faecal immunohistochemistry test. The neural network combined the FIT result with routinely collected data, including response to previous screening invitation; cases were confirmed using colonoscopy or alternative diagnostic test result within fourteen days. Of note, the authors included an equation for the resultant risk scores and provided details on how to manage applying the equation to new data with the need for standardisation of some of the variables. They did not undertake any external validation but did note the increased yield in positive colonoscopy results. This could have significant benefits where access to colonoscopy is an issue, including in the UK, where waiting lists have increased significantly, in part due to increasing demand [61].

Wang et al. [44] developed their neural network on a Taiwanese health insurance dataset. They noted the limitations of the data source, in particular, the lack of information on smoking status, alcohol intake, diet, and exercise. Interestingly, they used comorbidities as surrogates for these risk factors, for example, the use of smoking cessation clinic attendance and concurrent chronic obstructive pulmonary disease as confirmation of an individual as a “smoker”.

Missing or incomplete model information from the data extraction was commonly observed. Following the PRISMA guidelines for reporting on models was occasionally cited [51,52], but this was the exception rather than the rule. A lack of detail on handling missing data, inclusion, and exclusion criteria, and even the absence of calibration plots, was observed. Most studies included strengths and weaknesses of their approach, although comparison with other models was less frequent.

## 4. Discussion

### 4.1. Models of Note

Among the retrieved models, the most attractive is the one proposed by Kinar et al. (2016), which used an ensemble of decision (regression) trees. It has been evaluated in a wide range of situations and has even been subject to adjustment by variable addition. Datasets from Israel and the UK were used for the classification of CRC risk. The variables considered were limited to age, sex, and full blood count (FBC). Whilst age and sex data were complete, there was a significant lack of FBC from the UK THIN dataset (fewer reports per patient and more partial reports), yet this did not significantly affect the performance of the model when performance between the two datasets was compared. It has been further validated for a different UK dataset, the Clinical Practice Research Datalink (Birks et al., 2017). The model has also been evaluated in a wide range of situations and has been subject to adjustment by variable addition. The use of blood test result data is certainly interesting, but, as noted in their initial validation on UK data (THIN dataset), the availability of that information may be an issue. How well the UK dataset reflects standard practice in terms of the proportion of the population with an FBC and completeness of data is unclear. Assessing the level of availability of this information in other datasets would be a key part of adopting this approach. The study by Kop et al. [50] achieved good results using only age and gender, providing good evidence that applying this to other databank datasets, such as those in SAIL (Secure Anonymised Information Linkage), is a worthwhile initial step to take [56].

Another attractive approach utilises a neural network to stratify risk, with the clinical aim of informing screening test choice [52]. This study used the United States National Health Interview Survey dataset of personal health, a self-reported health questionnaire, to create a cohort of 525,394 (1269 cases) for the training set and 58,376 (140 cases) for the test set. The factors included had a strong correlation with colorectal cancer incidence but had to be mapped to the interval (0, 1) before using the neural network.

Smoking frequency had four categories and was assigned four values. The model was 10-fold cross-tested. The report misclassification rates were much lower than those obtained by following the USPSTF guidelines. Although it is not a screening model, the results suggest that stratification of the screening approach is possible and plausible. There are some limitations to the study, primarily including a lack of colonoscopy-confirmed cancer and the separation of the population into screened for training and not-screened for testing. The model performed better when family history was included, which is clinically unsurprising given a high correlation between family history and colorectal cancer diagnosis. There are some dataset-based limitations, such as the four-year cut-off between diagnosis and NHIS for discarding data and a lack of information on the CRC diagnosis. The model still requires validation and assessment of generalisability.

The model also includes comorbidity information and smoking status and identified diabetes as an important comorbid condition for colorectal cancer risk. However, assessment of diabetes severity, by creating an aggregated variable of diabetes complications as a sole predictor of colorectal cancer, performed poorly [43], but highlighted another potential iterative step in developing a prediction model. Adding a comorbidity measure (score) may be achievable with a range of datasets. For example, a modified Charlson index algorithm available within the SAIL (Secure Anonymised Information Linkage) gateway has been developed and used in non-cancer studies from this data [62] and takes account of missing comorbidity information, including the lack of routine availability of HIV status in the dataset. Several comorbidity measures were used in practice [63]. Even when only using administrative data [64,65], multiple versions of the Charlson index were available, and confirmation of the exact version used is necessary when reporting [66]. Indeed, choosing the correct comorbidity index requires significant consideration, despite the review by Stirland et al. [63], which has been subject to some criticisms, including regarding omitted algorithms [67]. The optimal algorithm for a comorbidity index remains unclear. The incorporation of morbidity indices in both CRC prediction and CC prognosis has been reviewed [68,69]; they were considered useful additions to existing algorithms [68], as well as increasing the likelihood of identifying emergency presentation and diagnosis [69].

It is worth noting that one study [41] developed a prospective model using a decision tree with only five features to predict adenomatous polyposis risk. Whilst the decision tree outperformed regression, this symptomatic population is very different to the general population or an asymptomatic at-risk population in its control-to-case ratio, so a similar performance in such a population is unlikely. The relatively small population also limits utilisation, but the results suggest that prospective studies are feasible. The authors noted that the decision tree did not utilise some established colorectal cancer risk factors, such as smoking and family history. Despite a decision tree providing “understandable” rules, it may remain difficult to understand clinically.

Finally, Kop et al. [50] discussed their development of a pre-processing pipeline. They identified key issues with data preparation, including ensuring that field entries were of uniform length, but, importantly, also noted limitations when excluding features that may not have a known association with the disease being predicted and the missed potential for discovering new predictors. This, perhaps, exemplifies the challenges and potential for machine-learning approaches in colorectal cancer risk prediction, the complexity of data preparation for model development, and the potential to discover yet undetected associations and risk factors for colorectal cancer development.

### 4.2. Main Discussion

From this scoping review focusing on machine-learning techniques, some common themes were identified. Whilst the limitations noted for models created with traditional methodology were also present for these ML-based models, differences in approach were also evident. Where ML-based models differ from traditional statistical techniques is in the use of extensive clinical data, including, but not limited to, comorbidity, medication, and blood testing, with generally larger numbers of variables considered, 1931 variables in one case [44]. The number utilised or included in the final model may be significantly reduced through feature selection techniques, or, in the case of neural network models, they can all be utilised, though their individual importance and use may be unclear due to the “black box” nature of such models. Model performance in terms of AUROC commonly exceeded 0.75, a value higher than that reported for models in the non-ML systematic reviews.

There is a need to overcome inertia in the clinical adoption of risk prediction and prognosis models in colorectal cancer. The reasons for poor utilisation are complex and not restricted to their performance or clinical acceptance [15]. A lack of randomised clinical trials of such models may not help their adoption, but research to develop novel ways of implementing such tools is taking place [70], with some requirement for early consideration of implementation strategies during predictive model development [71]. There have been few randomised controlled trials of algorithms, those that have been performed being limited to the assessment of medical images and related features [72].

Any colon-cancer prediction model needs to be able to accurately predict colorectal cancer risk, ideally at least three years prior to any potential diagnosis. The purposes of any cancer prediction model should include giving sufficient time to allow for modifications in risk or implementation of an appropriate monitoring strategy. The likelihood of risk modifications being applied in colon-cancer risk, or the level of benefit gained over a three-year period, is not known, although risk modification has been suggested to have the potential for significant impact [73]. A model should also be able to assess risk differences based on age at assessment and the likelihood of colorectal cancer during the highest risk period, the screening window, relating to age 50–75 years. That non-ML approaches predict risk with acceptable accuracy up to 15 years [14] from assessment is a challenge that none of the models reviewed are close to matching.

The prospective validation of ML models is uncommon, though, for CRC, the study by Wu et al. [41] shows that this is possible. Such validation is not without challenges, particularly in terms of data curation and the development of a pre-processing pipeline, as undertaken by Kop et al. [50]. Ensuring that the impact of feature differences over time are tested before continuing the use of even prospectively validated models is complex. By collecting data in advance of the first screening date, it may be possible to prospectively collect data to develop a prediction model. This could then be followed by further prospective validation. Whilst issues around take up of screening and, therefore, response to any data collection request are important considerations, routinely collected data, such as that found in a databank or registry, may provide an opportunity to apply this approach, including by use of linked resources, such as that being developed for the CORECT_R project [74]. CORECT-R aims to provide a single point for the linkage of data across the United Kingdom relating to cancer registry, hospital, treatment, patient experience and outcomes, with these being added to with other data, such as cancer waiting times and diagnostic imaging. Such “research ready” data provides opportunities to develop and test multiple models, including those utilising machine-learning techniques.

The choice of ML technique to adopt remains unclear from this review. Where multiple ML techniques have been used, tree-based models have tended to outperform other models, such as neural networks, and are more interpretable. The reason for this improved performance may be the ability to manage both linear and non-linear relationships between variables in the models [75]. Neural networks can consider a higher number of variables in CRC prediction. It is reasonable to include and compare multiple models, assess them and then identify the best performing models for further development. That there is no clear preferred technique may provide an opportunity for additional research. Comparing an interpretable and “black box” ML approach with similar performance for clinical use would be helpful. Is the ability to explain, both for the clinician and patient, more important than differences in performance? The choice of an ML technique may be determined by the type of data available, for example categorical versus continuous variables, to avoid manipulation of variables.

The generalisability of ML models is often mentioned and portability across countries and health systems suggested as the goal. Based on this review, this could inhibit the maximisation of a models’ performance within the environment for which it was specifically developed. The aim, therefore, should be to include data from all populations under consideration for which the model will be trained and tested. As shown in the various validation studies of the Kinar et al. [42] model, adaptation and extension were prevalent, without these amendments being checked in the original development population. Such amendments, whilst improving performance on the “new” population, lack an assessment of generalisability, nor do they consider what, how or where data is collected or data availability. This must be acknowledged by those making model adjustments. Importantly, the databanks for each country vary in their content and the level of population contributing to the data. The datelines and the data quality and availability across those datelines are also variable. Balancing data quantity and the contemporaneousness of information is very challenging. Changes in what data is collected and how tests are carried out are frequent. This means that those models utilising data collected over prolonged periods, which was common in the studies included in this current review, require a process to manage these changes and to equate differing values and normal ranges. Rather than being generalisable, a model should be responsive, identifying input data that does not match that of the training or test set. A model that can be adapted and revalidated with appropriate version control and data requirements may be more useful than a model developed in situation A and applied to situation B. The most generalisable model will be based on the fewest variables and involve the most regulated data collection, rather than those that may nuance the prediction of colon cancer within that specific population. What is required is that the model represent all parts of the population equally well, ensuring that the model can handle extremes of values within that population.

There has been criticism that using existing reporting guidelines for machine-learning approaches is flawed. To this end, extensions of the TRIPOD and PROBAST guidelines specifically for artificial-intelligence-based models are welcome [76,77]. It remains unclear whether application of the current TRIPOD and PROBAST guidelines impact negatively on the inclusion of ML models in previous systematic reviews. It is hoped that use of these AI-specific guidelines will increase the likelihood of machine-learning models being included in systematic reviews to allow assessment of a wider range of risk prediction models, regardless of the methodological approach.

Limitations of this review relate to definitions of routinely collected data and the blurring of machine-learning and traditional algorithm approaches. The initial literature search retrieved over 20,000 articles, the vast majority of which were excluded despite key words relating to machine learning in the search terms being used. This highlights the difficulty in identifying search terms and why machine-learning models were rarely identified in previous systematic reviews. Only 14 articles met the required inclusion criteria, which vindicated our scoping review approach. Heterogeneity in the health systems in which these models were developed was reflected in the variety of the variables that were considered and included in a final model. Routinely performed investigations and the subsequently collected data were as much a function of the health system as the selection process for their inclusion. Routine blood tests are not a feature in the United Kingdom but are performed elsewhere and, as such, the performance of a blood test in the United Kingdom may indicate that an illness or suspected illness is present and may not be considered as “routinely collected”. Furthermore, the search identified that an increasing volume of omics data is being collected, reflecting a change in “routine’ investigations as new technology is developed and the costs of performing such investigations fall. Therefore, it may not be possible to adequately define “routinely collected data” within a review that does not limit the health systems or locations considered.

## 5. Conclusions

This scoping review revealed the need to follow appropriate reporting guidelines, such as TRIPOD [78], during the planning, development, and application phases of any model. Robust assessment of performance is required with consideration given to the future prospective validation of any model that is developed.

Whilst external validation was rarely performed on the models reported, this can inform interpretation of generalisability. However, randomised controlled trials to confirm model performance may be more useful, particularly in terms of influencing clinical adoption.

Data-driven ML models were shown to perform well, generally performing at least as well as, or better than, traditional statistical models when compared with these.

ML techniques can provide a useful additional option to develop health models for CRC; however, it is unclear if they can be translated into clinical practice.

## Figures and Tables

**Figure 1 diagnostics-13-00301-f001:**
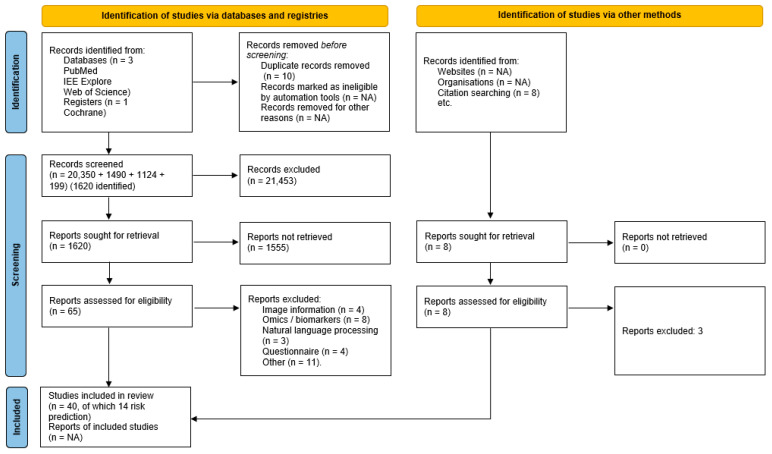
PRISMA flow diagram for systematic review. NA—not applicable.

**Table 1 diagnostics-13-00301-t001:** Summary of basic model details.

Lead Author	Machine-Learning Technique	Variables Assessed	Development (with Internal Validation)/Development with External Validation/Validation Only	Outcome	Case Definition
Wan, J.-J. 2019 [40]	Multiple ^1^	50 features assessed from endoscopic report, patient history and blood sample, age, gender, smoking history, and drinking history, endoscopic features (e.g., lesion location, polyp size, and no leaf) and blood attributes (e.g., white blood cells and haemoglobin)	Development No external validation	Prediction in 2 years (unclear)	Colonoscopy report confirmed neoplasia
Wu, H.-C. 2014 [41]	DT	20 parameters; demographic data (age, sex, body height, body weight, and body mass index), reasons for colonoscopy (bloody stool, abdominal pain, constipation, bowel habit change, anemia, tenesmus, positive faecal occult blood test (FOBT), colon polyp history, family history of CRC, or elevation in carcinoembryonic antigen level, and patient’s habits (smoking, drinking, betel nut chewing, or tea or coffee consumption)	Development No external validation	Adenoma presence Screening app ^2^	Colonoscopy pathology, adenomatous neoplasm
Kinar, Y. 2016 [42]	DT/RF	Sex, birth year and blood counts (20 parameters)	Development and validation	Prediction up to 2 years prior to diagnosis	CRC diagnosis in Israeli Cancer Registry/The Health Improvement Network (THIN) general practice database
Hsieh, M.-H. 2018 [43]	DNN	Demographic data, comorbidity and medication use; age, mean (SD) year, gender, urbanisation level, occupation, hypertension, hyperlipidemia, stroke, congestive heart failure, colorectal polyps, obesity, chronic obstructive pulmonary disease, coronary artery disease, asthma, smoking, inflammatory bowel disease, irritable bowel syndrome, alcohol-related illness, chronic kidney disease, diabetes complication (aDCIs component), retinopathy, nephropathy, neuropathy, cerebrovascular, cardiovascular, peripheral vascular disease, metabolic, mean aDCIs score onset, end of follow-up, statins, insulin, sulfonylureas, metformin, thiazolinedione, other antidiabetic drugs and mean follow-up for endpoint. Selection of input features ND, the abstract states “All the available possible risk factors for CRC were also included in the analyses”	Development No external validation	Prediction, timeframe unclear	CRC diagnosis recorded in National Health Insurance Research Database (NHIRD)
Wang, Y.-H. 2019 [44]	CNN	Comorbidity and medication use with age and sex	Development No external validation	Prediction within 3 years	CRC diagnosis recorded in Registry for Catastrophic Illness Patient
Hornbrook, M.C. 2017 [45]	DT/RF	Gender, age, and blood count (number of parameters—at least one)	Validation of Kinar 2016	Prediction within 6 months to 1 year	CRC diagnosis in Kaiser Permanente Tumor Registry
Birks, J. 2017 [46]	DT/RF	Gender, age, and blood count (20 parameters)	Validation of Kinar 2016	Prediction in 18–24 months	CRC diagnosis recorded in Clinical Practice Research Datalink (CPRD)
Kinar, Y. 2017 [47]	DT/RF	Gender, age, and blood count (20 parameters) at 2 timepoints	Evaluation of Kinar 2016	Prediction within 6 months	CRC diagnosis in Israeli Cancer Registry
Goshen, R. 2017 [48]	Linear Regression	Gender, age, blood count, liver function, metabolic blood tests, and iron, folic acid, and vitamin B12 levels.	Development, extension of Kinar 2016	Prediction within 6 months	CC diagnosis in Israeli Cancer Registry
Hilsden, R.J. 2018 [49]	DT/RF	Gender, age, and blood count; one or more from haemoglobin, haematocrit, mean corpuscular volume, mean corpuscular haemoglobin, mean corpuscular haemoglobin concentration, red blood cell count, red blood cell distribution width, white blood cell count, platelets, % neutrophils, lymphocytes, monocytes, eosinophils, and basophils—up to 15 variables.	Validation of Kinar 2016	Prediction within 12 months	Colonoscopy result of CRC or high-risk polyp.
Kop, R. 2016 [50]	RF/CART	Age and gender with medical-record-based data for consultations, medication, referral, diagnoses and lab. test results; drugs for constipation; iron deficiency anaemia; lipid modifying agents (s); drugs for constipation; age; drugs for acid-related disorders (s); drugs for constipation; diabetes non-insulin-dependent; abdominal pain/cramps general; diabetes non-insulin-dependent (s); diabetes non-insulin-dependent; beta-blocking agents (s) ; drugs for constipation; hypertension uncomplicated (s) ; hypertension uncomplicated; agents acting on the renin–angiotensin system (s); drugs for constipation; diuretics; flu vaccination (a); agents acting on the renin–angiotensin system (s); Antithrombotic agents; abdominal pain localized other; general consult (s); agents acting on the renin–angiotensin system (s) ; drugs for acid-related disorders; agents acting on the renin–angiotensin system	Development No external validation	ND	CRC diagnosis in general practice dataset
Cooper, J.A. 2018 [51]	ANN	Age, sex, deprivation index, screening history, FIT test result	Development No external validation	Predict CRC/high risk adenoma versus none following FIT	Bowel cancer screening database record of CRC or advanced adenoma
Nartowt, B.J. 2019 [52]	ANN	Health questionnaire responses; current or cancer age, hypertension, number of first-degree relatives with CRC (NHIS years 2000, 2005, 2010, and 2015 only), coronary heart disease, pooled heart conditions, myocardial infarction, diabetes (non-gestational), heart condition/disease, vigorous exercise frequency, angina pectoris, ulcer (stomach, duodenal, peptic), Hispanic ethnicity, stroke, emphysema, American Indian, African American, other, or multiple race, sex (male), body-mass index, smoking frequency	Development No external validation	Risk score generation	Confirmed colorectal cancer (any) in NHIS dataset
Shi, Q. 2019 [53]	CART	Rate of albumin to globulin, albumin, alanine transaminase, aspartate transaminase, percent basophils, calcium, creatinine, percent eosinophils, glucose, hematocrit, high-density lipoprotein-cholesterol, haemoglobin, potassium, low-density lipoprotein-cholesterol, percent lymphocytes, mean corpuscular haemoglobin, mean corpuscular haemoglobin concentration, mean corpuscular volume, percent monocytes, mean platelet volume, percent neutrophils, phosphorus, platelet large cell ratio, plateletcrit, platelet distribution width, platelet, red blood count, variable coefficient of red blood cell distribution width, standard deviation of red blood cell distribution width, total bilirubin, total cholesterol, triglyceride, total protein, uric acid, white blood count Final model used: age, albumin, haematocrit, % lymphocytes	Development, internal validation.	Risk score generation	Confirmed colorectal cancer

(^1^) Multiple machine-learning methods assessed; support vector machine (SVM), k-nearest neighbours (KNN), ensembles for boosting, random forest, convolutional neural network, recurrent neural network, recursive neural network. (^2^) A screening app for Android smartphones was derived from the model. Abbreviations: CRC—colorectal cancer, CC—colon cancer, ND—not described, DT—decision tree, RF—random forest, DNN—deep neural network, CNN—convolutional neural network, ANN—artificial neural network, CART—classification and regression tree.

**Table 2 diagnostics-13-00301-t002:** Summary of model purpose, study population and data source(s).

Lead Author	Age Range (Years)	Sample Size	Data Source
Wan, J.-J. 2019 [40]	ND	ND	China, Jiangsu Provincial Hospital of Traditional Chinese Medicine
Wu, H.-C. 2014 [41]	Cases 21–80 Controls 34–80	225 (97 cases)	Taiwan, single centre, unnamed
Kinar, Y. 2016 [42]	Cases > 40 Controls 50–75	Complex (*)	UK, The Health Improvement Network (THIN) database/Israel, Maccabi Health Services
Hsieh, M.-H. 2018 [43]	>20	1,315,899 train 337,410 test (14,867 cases)	Taiwan, subset of National Health Insurance Research Database (NHIRD) and Longitudinal Cohort of Diabetes Patients (LHDP)
Wang, Y.-H. 2019 [44]	>20	47,967 controls, 10,185 cases	Taiwan, National Health Insurance Research Database (NHIRD)
Hornbrook, M.C. 2017 [45]	40–89	10,008 (900 cases)	USA, Single Institution Registry, Kaiser Permanente Northwest Region
Birks, J. 2017 [46]	>40	2,220,108 (25,430 cases)	UK, Clinical Practice Research Datalink (CPRD)
Kinar, Y. 2017 [47]	50–75	112,584 (133 cases)	Israel, Maccabi Health Services and Israeli Cancer Registry
Goshen, R. 2017 [48]	40–75	105,067 (2294 cases) 1755 cases and 54,730 matched controls in study	Israel, Maccabi Health Services and Israeli Cancer Registry
Hilsden, R.J. 2018 [49]	50–75	17,676 (60 CRC, 1104 high risk polyps—cases) (screened)	Canada, Alberta Health Services Forzani and MacPhail Colon Cancer Screening Centre in Calgary, AB, Canada by linking the Centre’s electronic medical records with provincial laboratory data.
Kop, R. 2016 [50]	≥30	263,879 (1292 cases)	Netherlands, Julius General Practitioners’ Network, Utrecht; Academic Network of General Practice, VU University Medical Center Amsterdam (ANH VUmc); Leiden General Practitioner Registration Network RNUH-LEO, LUMC, Leiden.
Cooper, J.A. 2018 [51]	60–74	1810 (548 cases—cancer, high or intermediate risk polyps) (screened patients only)	UK, two regional cancer screening hubs (NHS Bowel Screening)
Nartowt, B.J. 2019 [52]	18–85 (18–49, 50–75)	525,394 train (1269 cases) 58,376 test (140 cases)	USA (National Health Interview Survey) (**)
Shi, Q. 2019 [53]	ND	PUCH 81,310 (4211 cases 70:30 train:test PUSH 57,235 (436 cases) 80:20 train:test Validated on PUCH test set	China: Peking University Cancer Hospital (PUCH) and Peking University Shougang Hospital (PUSH)

* Sample size for Kinar et al., 2016 included populations from two countries with the Israeli population split into a derivation set and validation set and a UK dataset for validation only. ** A dataset curated by the Center for Diseases Control. Abbreviations: CRC—colorectal cancer, CC—colon cancer, ND—not described.

**Table 3 diagnostics-13-00301-t003:** Summary of Model Performance Measures—“best” model where multiple models tested.

Reference	Model Type	Model Performance Measure											Misc
		AUROC	Sensitivity	Specificity	PPV	NPV	FPR	FNR	Accuracy	Precision	F1 Score	OR	
Wan, J.-J. 2019 [40]	Neural network (ECP)		0.6					0.7321	0.8148	0.8571	0.7059		
Wu, H.-C. 2014 [41]	Decision tree	0.937	0.825	0.922			0.078	0.175				26 (+/− 5)	
Kinar, Y. 2016 [42]	Combined	0.82 (+/− 0.01)		0.88 (+/− 0.02)			0.005					40 (+/− 6)	FPR and OR at 50% case detection
		0.81		0.94 (+/− 0.01)									
Hsieh, M.-H. 2018 [43]	Neural Network	0.7 (0.674–0.727)	0.886							0.98	0.929		Test set values
Wang, Y.-H. 2019 [44]	Neural network	0.922 (SD 0.0004)	0.837	0.867	0.532								
* Hornbrook, M.C. 2017 [45]	Combined	0.8 (0.79–0.82)										34.7 (28.9–40.4)	OR at 99% specificity
* Birks, J. 2017 [46]	Combined	0.776 (0.771–0.781)	3.91 (3.4–4.48)	82.73 (82.68–82.78)	0.088	0.996							
* Kinar, Y. 2017 [47]	Combined		0.173									21.8 (13.8–34.2)	At 1% percentile of scores, yield 2.1%
** Goshen, R. 2017 [48]	Logistic regression		0.31	0.95	0.073								males
			0.24	0.95	0.042								females
* Hilsden, R.J. 2018 [49]	Combined		0.081 (0.064–0.098)									5.1 (2.3–8.9)	At 95% specificity, OR versus no findings
Kop, R. 2016 [50]	Logistic regression	0.891	0.642							0.03	0.058		
Cooper, J.A. 2018 [51]	ANN	0.686 (0.659–0.712)	0.3515	0.8557	0.5147	0.7519							10.66% CRC detection rate
Nartowt, B.J. 2019 [52]	Neural net	0.80 (+/− 0.05)	0.57 (+/− 0.03)	0.89 (+/− 0.02)									NPV and PPV in abstract
Shi, Q. 2019 [53]	CART	0.88 (0.87–0.90)	0.622 (0.581–0.662)										Sensitivity at 99% specificity

* Validation studies, ** extension study. Abbreviations: ANN—artificial neural network, CART—classification and regression tree, combined—gradient boosted model and random forest ensemble, AUROC—area under the receiver operator curve, PPV—positive predictive value, NPV—negative predictive value, FPR—false positive rate, FNR—false negative rate, OR—odds ratio.

## Data Availability

Not applicable.

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
