# Peer review of "Machine Learning in Colorectal Cancer Risk Prediction from Routinely Collected Data: A Review"

_diagnostics, 2023, doi:10.3390/diagnostics13020301_

Round 1

Reviewer 1 Report

The manuscripts conduct quite extensive literature study aims at 1) identifying machine learning models, 2) assessing their methodology as well as 3) comparing their performance with that found in previous reviews. Yet, rather than just reporting previous study applying machine learning for Colorectal Cancer Risk Prediction from 2 Routinely Collected Data, the study should reveals potential questions that represents the gap in the body of knowledge dealing with the topic.

Author Response

We are grateful to the editor and anonymous reviewers for their constructive comments and suggestions which have helped us to further improve the quality of this manuscript. We have fully
addressed these comments and suggestions. Below, we summarise how we have responded to them.

“Yet, rather than just reporting previous study applying machine learning for Colorectal Cancer Risk Prediction from 2 Routinely Collected Data, the study should reveals potential questions that represents the gap in the body of knowledge dealing with the topic.”

Response: The discussion has been updated to address this.

Revision Location : Discussion.

Reviewer 2 Report

1. Starting from the title: It mentions scoping review but in the abstract first line, it mentions systematic reviews of risk prediction models of the other models. Systematic review would have already provided the risk prediction models and further details

2. Abstract does not mention the how many publications. Similarly, "A literature search was performed for publications that included at least one machine learning model from four databases." Now what is point here to mention scoping review ? when it is already systematic review which is probably a copy of the previous one. Further, the author themselves mention "This review found similar methodological problems to previous reviews, although model performance was better". So why this study is conducted then ?

3. Prior literature is missing, although some provided in introduction but following a broader approach

4. Inclusion criteria; full text availability, English Language, articles related to colorectal or colon cancer, published between 2010 and 2019 inclusive, Why not older? or why not up to 2022. Why 3 years old

5. in results it is not clear that how author reached up from 20,350 articles to 14 publications only. Also, that is too less for a systematic review

6.  In summary table, authors are assessing 50 features or sometime 20 parameters. Why this criteria was not limited to have relevant models. Every single variable approach could be different. 

7. Not sure that what authors meant by critical appraisal. Already in prior sections, same repetitive information is provided. This section is like a literature author by author

8. There isnt anything valuable found in the main discussion section 4.2> There is nothing different from the prior discussion sections. 

9. Authors just pasted the information as from other articles in this article, there is no difference and value in the conducted research. It is simple literature review. Even the process for the articles, coding, and reaching the output the systematic review is missing. 

Author Response

1. "Starting from the title: It mentions scoping review but in the abstract first line, it mentions systematic reviews of risk prediction models of the other models. Systematic review would have already provided the risk prediction models and further details”
Response: The mention of systematic reviews was intended to frame the approach taken for our paper.
We have altered the text to reflect that.
Revision Location : Abstract, lines 10– 13.
2.“Abstract does not mention the how many publications”
Response: This has been rectified and added to the abstract.
Revision Location : Abstract, line 17.
3.“Similarly, "A literature search was performed for publications that included at least one machine
learning model from four databases." Now what is point here to mention scoping review ?”
Response: This has been rectified and text altered accordingly.
Revision Location : Abstract, lines 19 - 20.
4.“when it is already systematic review which is probably a copy of the previous one. Further, the author themselves mention "This review found similar methodological problems to previous reviews, although model performance was better". So why this study is conducted then ?”
Response: The absence of machine learning models from the systematic reviews could not be explained from the systematic reviews. It was unclear if their exclusion was systematic from inclusion and exclusion criteria and if not, whether the methodological concerns found in the reviews of non-machine learning models would persist. Text has been altered to emphasise this.
Revision Location : Abstract, lines 12 – 13.
5.“Prior literature is missing, although some provided in introduction but following a broader approach”
Response: Additional literature added to highlight the expansion of features under consideration and their inclusion and consideration in traditional models.
Revision Location : Introduction, lines 55 – 60.
6. “Inclusion criteria; full text availability, English Language, articles related to colorectal or colon cancer, published between 2010 and 2019 inclusive, Why not older? or why not up to 2022. Why 3 years old”
Response: We have clarified that the publication date window was extended to include publications from 2021.
Revision Location : Eligibility Criteria 2.1, lines 118 – 119.
7. “in results it is not clear that how author reached up from 20,350 articles to 14 publications only. Also, that is too less for a systematic review”
Response: We apologise for the inadvertent omission of the PRISMA diagram (figure 1) which shows the article processing in more detail. We agree regarding a systematic review approach requiring more articles and our scoping review approach was undertaken on the basis that it was unclear how many models / publications might be identified, an explanation has been added to the text.
Revision Location : Article Selection 3.1, diagram added to line 206. Materials and Methods 2, lines
104 – 111.
8. “In summary table, authors are assessing 50 features or sometime 20 parameters. Why this criteria was not limited to have relevant models. Every single variable approach could be different.”
Response: Using the scoping review approach allowed us to identify that identified models had a wide range in the number and variety of features included in the development of and indeed, the final models themselves. This has been elaborated on in the eligibility criteria.
Revision Location : Eligibility criteria, lines 121 – 124.
9. “Not sure that what authors meant by critical appraisal. Already in prior sections, same repetitive information is provided. This section is like a literature author by author”
Response: Models were benchmarked against the CHARM checklist, we have changed the terminology to clarify this.
Revision Location : Section 3.5
10. “There isnt anything valuable found in the main discussion section 4.2> There is nothing different from the prior discussion sections.”
Response: We have removed potential overlap between the sections in the discussion and focused on remaining knowledge gaps.
Revision Location : Section 4.2, lines 432 – 441, 461 – 465, 477 – 519, 548 – 553.
11. “Authors just pasted the information as from other articles in this article, there is no difference and value in the conducted research. It is simple literature review. Even the process for the articles, coding, and reaching the output the systematic review is missing.”
Response: The previous clarifications should redress this concern.

Round 2

Reviewer 2 Report

Abstract is adequate now

Prior literature is missing: still missing

“in results it is not clear that how author reached up from 20,350 articles to 14 publications only. Also, that is too less for a systematic review”: still missing
